# Advanced MDCT assessment of abdominal aortic wall integrity and morphometry in the Saudi cohort: A single-centre cross-sectional study

**Mohammad Ahmad Mostafa Alloush**[1], **Mazin Babikir Hassib**[2], **Husain Alturkistani**[3], **Rafat S. Mohtasib**[1,4], **Rashed Ali Alhamed**[5], **Reham Mukhlid Almutairi**[5], **Mohammed J. Alsaadi**[6]*

**1** Department of Medical School, Alfaisal University, Riyadh, Saudi Arabia, **2** University of Hail, College of Applied Medical Sciences, Diagnostic Radiology Department, Hail, Saudi Arabia, **3** Department of Radiology & Medical Imaging, College of Medicine - King Saud University, Riyadh, Saudi Arabia, **4** King Faisal Specialist Hospital & Research Centre, Riyadh, Saudi Arabia, **5** Medical Imaging Administration, King Fahad Medical City, Riyadh, Saudi Arabia, **6** Radiology and Medical Imaging Department, College of Applied Medical Sciences in Al-Kharj, Prince Sattam Bin Abdulaziz University, Al-Kharj, Saudi Arabia

* m.alsaadi@psau.edu.sa

## Abstract

### Background

Accurate reference values for abdominal aortic dimensions and wall thickness are crucial for the early detection and management of vascular diseases, particularly abdominal aortic aneurysms (AAAs). However, there is a lack of population-specific normative data for the Saudi population.

### Methods

A cross-sectional study was conducted on 347 adults [111 males, 236 females] aged 20–80 at King Fahad Medical City. Multidetector computed tomography (MDCT) was utilised to measure the abdominal aortic lumen area, diameter, and wall thickness at three standard anatomical levels. Pixel-based segmentation and image analysis using MATLAB allowed for precise quantification of wall thickness. Multiple linear regression was employed to assess associations with age, gender, and BMI.

### Results

The average diameter of the aorta was 1.87 cm, the average wall thickness was 1.6 mm, and the average lumen area was 3.01 cm². Males generally had thicker aortic walls and larger dimensions than females, particularly in younger age groups. However, these gender differences became less pronounced with older age and increased BMI. A higher BMI was strongly linked to thicker aortic walls ($p < 0.001$) but was also associated with a reduced lumen area ($p < 0.001$). Importantly, aortic diameter did not significantly influence wall thickness.

**Data availability statement:** All relevant data are within the paper and its Supporting information files.

**Funding:** The author(s) received no specific funding for this work.

**Competing interests:** None.

## Conclusion

This study provides essential normative data for abdominal aortic measurements in the Saudi population, highlighting age and BMI as significant influencers of aortic morphology. The results support the adoption of population-specific diagnostic criteria and demonstrate the utility of advanced MDCT-based measurement techniques in vascular assessment.

## Introduction

The abdominal aorta, the primary conduit for oxygenated blood to the lower body, originates at the aortic hiatus of the diaphragm and bifurcates into the common iliac arteries at the level of the fourth lumbar vertebra [1]. Its structural integrity is crucial for systemic perfusion and is influenced by physiological factors, including age, sex, and body composition [2]. The aortic wall is anatomically composed of three layers: the intima, media, and adventitia—each uniquely contributing to vascular function, elasticity, and support [3]. Clinically, the normal diameter of the abdominal aorta is less than 3 cm, but this can vary according to demographic and physiological factors [4]. Abnormal dilation, such as abdominal aortic aneurysms (AAA), is generally defined as having a diameter exceeding 3 cm. AAAs are often asymptomatic and are frequently identified incidentally during imaging for unrelated conditions. The clinical risk of rupture, which is associated with high morbidity and mortality, emphasises the necessity for accurate and timely diagnosis [5,6].

Recent evidence suggests that, beyond diameter, aortic wall thickness may also serve as a crucial marker for vascular pathology. Studies indicate that ruptured aneurysms often show thicker walls than their unruptured counterparts, highlighting the diagnostic value of assessing luminal and wall parameters [7,8]. Despite advancements in imaging technology, the absence of population-specific normative data, particularly among Saudi populations, limits diagnostic accuracy and risk stratification. While modalities such as ultrasound, CT, and MRI are employed for aortic evaluation, multi-detector computed tomography (MDCT) provides superior spatial resolution, facilitating a detailed assessment of the aortic wall and lumen [9]. However, current diagnostic protocols often rely on one-dimensional measurements and lack regional calibration, which can lead to misclassification or delayed intervention [10].

The Saudi population remains underrepresented in vascular imaging studies, despite a rising prevalence of cardiovascular risk factors such as obesity and hypertension [11–13]. Despite increasing rates of cardiovascular risk factors in the Gulf region, the Saudi population remains underrepresented in vascular imaging studies [13]. This data gap limits the development of effective screening protocols calibrated for local populations. Therefore, it is essential to establish normative abdominal aortic measurements, including wall thickness, within this population to develop effective screening, monitoring, and intervention strategies. This study utilises MDCT to determine baseline values for abdominal aortic diameter, wall thickness, and lumen area in an adult Saudi cohort, by correlating these parameters with age, gender, and body

mass index (BMI). By establishing normative values for abdominal aortic dimensions in a healthy Saudi population, this research aims to contribute to a population-specific diagnostic framework. This involves refining thresholds for the early detection of vascular abnormalities, particularly abdominal aortic aneurysms (AAAs), and enabling improved risk stratification tailored to regional demographic profiles.

## Materials and methods

### Study design and population

This retrospective, cross-sectional analytical study was conducted at King Fahad Medical City (KFMC) in Riyadh, Saudi Arabia. The data of participants who underwent contrast-enhanced multidetector computed tomography (MDCT) of the abdomen at KFMC from 2023 to 2024 were included. The data was retrieved from the PACS on October 10, 2024. The study included 347 adult participants [111 males and 236 females] aged between 20 and 80. Individuals with known cardiovascular or systemic diseases or a prior history of vascular interventions were excluded to ensure that the measurements reflected baseline physiological anatomy. Systemic diseases were defined as including autoimmune, inflammatory, and connective tissue disorders known to affect vascular morphology. The included MDCT scans were performed for non-vascular indications such as abdominal pain or incidental evaluation, and patient records were reviewed to confirm the absence of known vascular pathology. Ethical approval for this study was obtained from the Institutional Review Board (IRB) of King Fahad Medical City (IRB Log Number: 23–411), in accordance with the International Council for Harmonisation Good Clinical Practice (ICH-GCP) guidelines. As the study was retrospective in nature and involved the analysis of de-identified data, the requirement for informed consent was waived by the IRB.

### Imaging protocol

The data of participants who underwent contrast-enhanced multidetector computed tomography (MDCT) of the abdomen at the KFMC archiving system were included. A General Electric CT system was used, with image acquisition conducted at a slice thickness of 2.5 mm and an 80% spacing. Multiphase abdominal CT scans were obtained, and arterial-phase images were reconstructed into axial, sagittal, and coronal planes using multiplanar reformation (MPR) techniques.

### Measurement methodology

Post-processing was conducted using a DICOM viewer and MATLAB software for precise measurements. The following parameters were evaluated: the Aortic Lumen area was measured at three standard anatomical levels (pre-mesenteric, mesenteric, and post-mesenteric) by employing polygonal region-of-interest (ROI) tools on axial images (Fig 1). The study focused on three essential parameters—diameter, wall thickness, and lumen area—because of their clinical importance, measurement consistency, and frequent reporting in the literature. This choice enables direct comparison with global datasets and emphasises metrics most significant for diagnosing AAA.

The average of three readings was utilised for analysis. A MATLAB-based segmentation method was used to calculate the wall thicknesses by measuring of the pixel-to-pixel distance between the inner and outer aortic boundaries on axial images (Fig 2). The MATLAB-based image segmentation process used edge detection algorithms to delineate the inner and outer vessel boundaries on axial CT slices. Manual annotation by two independent radiologists was used to validate boundary detection, and inter-observer variability was assessed using intraclass correlation coefficients (ICC). The resulting ICC value of 0.89 indicated strong agreement between observers, supporting the reliability of the segmentation method. This methodological approach provides high precision. It represents a novel contribution to quantitative aortic imaging. Manual annotation ensured accuracy and consistency. The Body Mass Index (BMI) was calculated using the standard formula: BMI = weight (kg)/ height² (m²), following WHO BMI classification guidelines [14].

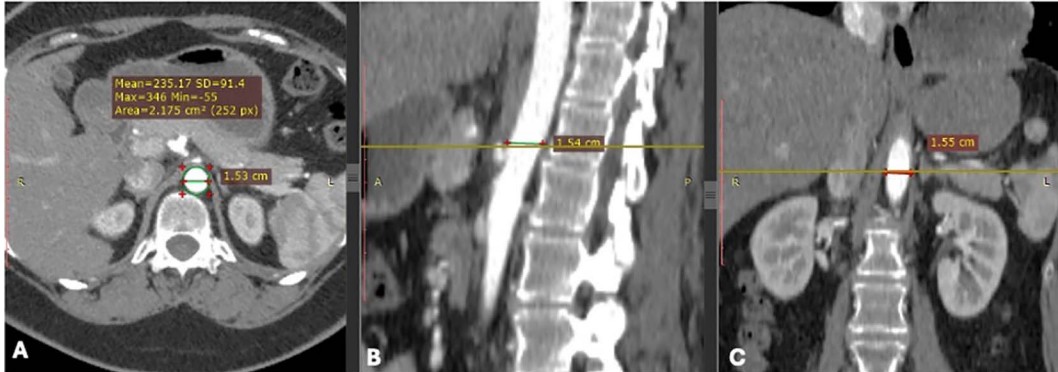

**Fig 1. CTA visualising the abdominal aorta; (A) shows an axial section illustrating the measurement of abdominal aorta diameters using polygons (ROIs).** (B) displays a sagittal section measuring the length of the abdominal aorta. (C) demonstrates a coronal section measuring the width of the abdominal aorta.

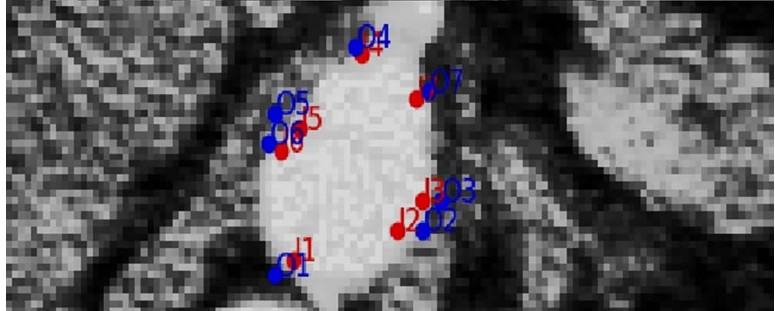

**Fig 2. The region of interest in CTA used to measure aortic wall thickness.** The red label designates the inner point, while the blue label denotes the outer point.

## Variables and statistical analysis

Collected variables included age, gender, height, weight, BMI, aortic lumen area (cm²), wall thickness (mm), and aortic diameter (cm) from axial, sagittal, and coronal views. Statistical analysis was performed using multiple linear regression to examine the associations between aortic wall thickness and predictors, including age, gender, BMI, aortic diameter, and lumen area. A p-value of <0.05 was considered statistically significant.

## Results

A total of 347 adult participants [111 (32%) male and 236 (68%) female] were included in this study. The participants ranged in age from 20 to 80 years, with a mean age of $50 \pm 15$ years. Key anthropometric and imaging-derived measurements are summarised in Table 1. The mean body mass index (BMI) was $27.5 \pm 5.3$ kg/m², indicating a general trend towards being overweight. The mean abdominal aortic diameter was $1.87 \pm 0.26$ cm, the mean aortic wall thickness was $1.6 \pm 0.26$ mm, and the average lumen area was $3.07 \pm 0.78$ cm². The average abdominal aortic diameters were nearly identical across imaging planes: Axial: $1.87 \pm 0.26$ cm, Coronal: $1.87 \pm 0.26$ cm, and Sagittal: $1.81 \pm 0.26$ cm. This consistency underscores the reliability of cross-sectional diameter measurements across standard views.

**Table 1. Descriptive Statistics of Participant Characteristics and Aorta Measurements.**

| Variable | Count | Minimum | Maximum | Mean | ± SD |
|---|---|---|---|---|---|
| **Participant Characteristics** | | | | | |
| Age (years) | 347 | 20.00 | 80.00 | 50.00 | 15.00 |
| Height (m) | 347 | 1.35 | 1.98 | 1.72 | 0.12 |
| Weight (kg) | 347 | 45.00 | 149.00 | 78.41 | 17.76 |
| BMI (kg/m²) | 347 | 16.13 | 40.20 | 27.50 | 5.30 |
| **Aorta Measurements** | | | | | |
| Lumen Width (cm²) | 347 | 1.35 | 4.97 | 3.07 | 0.78 |
| Wall Thickness (mm) | 347 | 0.70 | 2.90 | 1.60 | 0.26 |
| Axial Section Diameter (cm) | 347 | 1.30 | 2.44 | 1.87 | 0.26 |
| Coronal Section Diameter (cm) | 347 | 1.26 | 2.44 | 1.87 | 0.26 |
| Sagittal Section Length (cm) | 347 | 1.07 | 2.49 | 1.81 | 0.26 |

*Note*: Aorta measurements based on imaging data (N = 347). SD = Standard Deviation.

## Linear regression analysis

Multiple linear regression was performed to evaluate the effect of demographic and anatomical variables on wall thickness (Table 2): Gender was significantly associated with wall thickness (p = 0.017), with males showing thicker aortic walls. Age was inversely associated with wall thickness [β = −0.0099, p = 0.005], indicating a decrease in wall thickness with increasing age (Fig 3). BMI showed a strong positive correlation [β = 0.0509, p < 0.001], suggesting that higher BMI is associated with increased wall thickness (Fig 4). Lumen area was negatively correlated with wall thickness [β = −0.2116, p < 0.001], Fig 5. Aortic diameter did not significantly predict wall thickness (p = 0.63), Fig 6. The adjusted R² for the model was 0.62. All variance inflation factors (VIFs) were below 2.5, indicating low multicollinearity. Standard errors for each coefficient are reported in Table 2. Residual plots were inspected and found to approximate normality, justifying the use of linear regression assumptions.

Males exhibited larger aortic diameters and thicker walls in younger age groups (<50 years). With advancing age (>50 years), gender-based differences in aortic measurements diminished, demonstrating convergence in median values and variability. This trend was consistent across aortic diameter, lumen area, and wall thickness. The effects of BMI categories on aortic characteristics were analysed. In the normal and overweight BMI groups, males exhibited slightly greater wall thickness and lumen areas than females. In the obese group, the gender gap diminished, with nearly identical values across both sexes. The findings confirm that BMI exerts a more substantial influence on wall thickness than gender, particularly in the higher BMI categories.

**Table 2. Multiple Regression Results for Wall Thickness (mm).**

| Predictor | Coefficient | SE | t-value | 95% CI lower | 95% CI upper | p-value |
|---|---|---|---|---|---|---|
| Intercept | 1.440 | 0.100 | 14.400 | 1.242 | 1.638 | <0.001* |
| Gender (Female vs. Male) | −0.150 | 0.055 | −2.730 | −0.259 | −0.041 | 0.017* |
| Age (years) | −0.010 | 0.004 | −2.829 | −0.017 | −0.003 | 0.005* |
| BMI (kg/m²) | 0.051 | 0.008 | 6.284 | 0.035 | 0.067 | <0.001* |
| Aorta Diameter (cm) | −0.005 | 0.011 | −0.476 | −0.026 | 0.016 | 0.634 |
| Lumen Width (cm²) | −0.212 | 0.028 | −7.588 | −0.267 | −0.157 | <0.001* |

*Note*: SE = Standard Error, CI = Confidence Interval. *p < 0.05 indicates statistical significance.

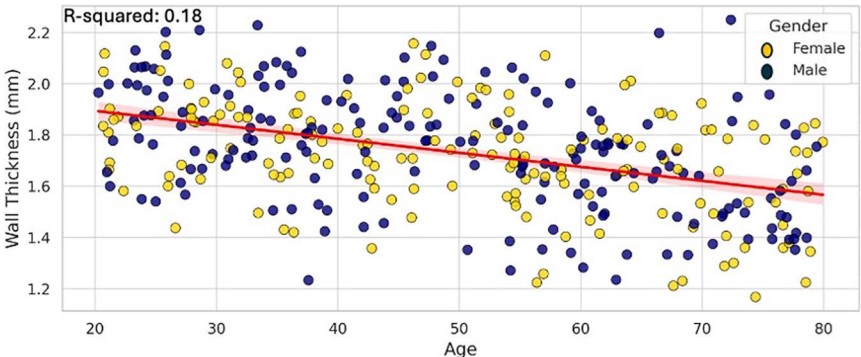

**Fig 3. The linear regression, with a 95% confidence interval around the line (shown as a shaded band) in the scatter plot, demonstrates the relationship between age and wall thickness, differentiated by gender.**

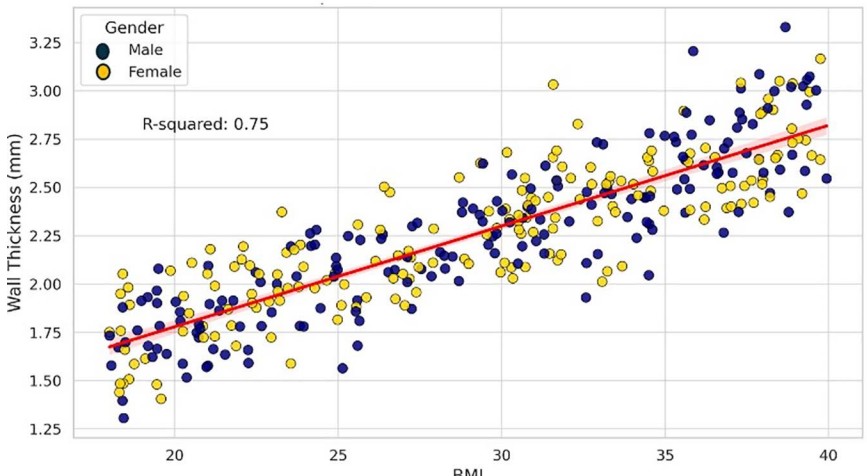

**Fig 4. The linear regression scatter plot illustrates the relationship between BMI and wall thickness, categorised by gender, with a 95% confidence interval surrounding the line (represented by a shaded band).**

Fig 7 presents a 3D scatter plot that integrates age, gender, BMI, wall thickness, and lumen width. The visualisation supports the regression findings: wall thickness decreases with age and increases with BMI. Gender differences are significant in early adulthood but become negligible in older or obese individuals. Lumen area and wall thickness are inversely related across all groups.

## Discussion

This study employed Multi-Detector Computed Tomography (MDCT) to establish normative measurements of abdominal aortic dimensions, including diameter, wall thickness, and lumen area, in healthy Saudi adults. The results provide population-specific reference values, addressing a critical gap in regional anatomical and clinical data. By analysing the associations between demographic variables—age, gender, and body mass index (BMI)—and aortic dimensions, this research enhances diagnostic accuracy and facilitates improved clinical decision-making. Our study builds upon previous large-scale investigations, such as the Framingham Heart Study and the Copenhagen General Population Study, by providing normative aortic measurements specific to the Saudi population [15,16]. While these earlier studies offered broad

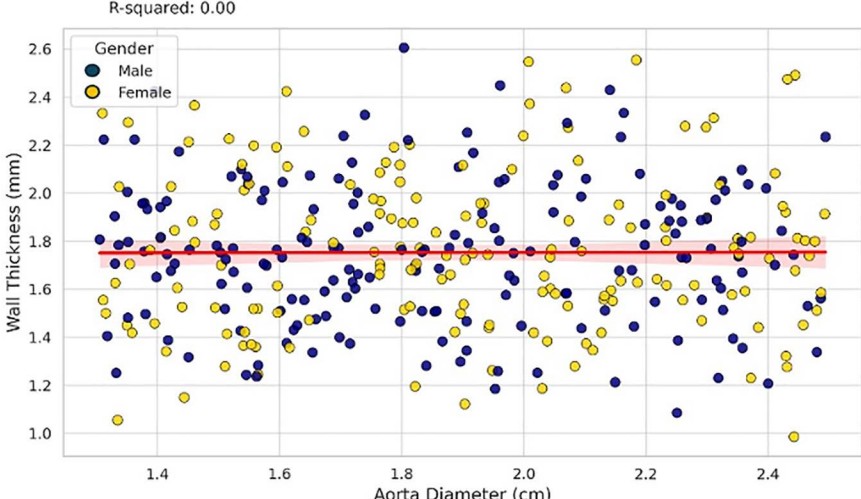

**Fig 5. A flat regression line with a visible 95% confidence interval regression plot shows the relationship between aorta diameter (in cm) and wall thickness (in mm), distinguished by gender.**

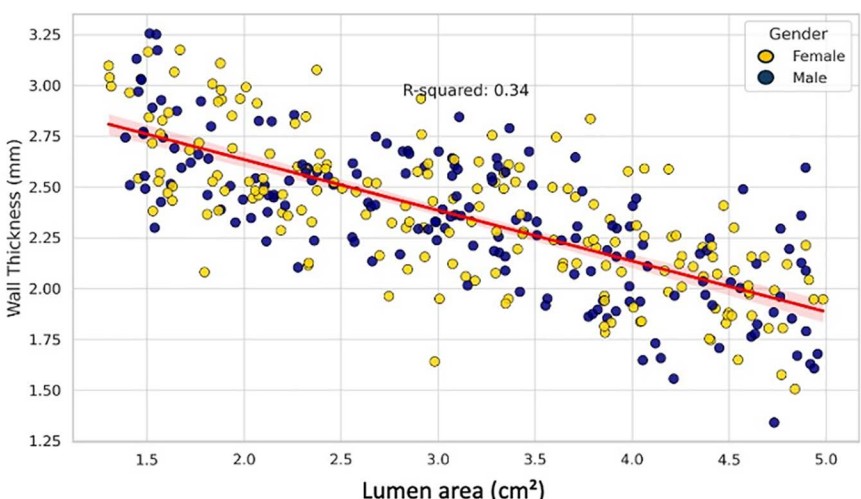

**Fig 6. The scatter plot of linear regression with 95% confidence interval shows the relationship between lumen area (cm²) and wall thickness (mm), distinguished by gender.**

population-level data, they did not address regional anatomical variations or use pixel-based wall thickness segmentation. Compared to Framingham values, our cohort exhibited slightly smaller mean aortic diameters and thicker wall profiles in younger males, suggesting possible ethnic or environmental influences. These differences emphasise the importance of developing localisation reference values for more accurate diagnostic application.

Our analysis identified significant relationships between demographic factors and aortic measurements. A notable finding was the negative correlation between age and aortic wall thickness, confirmed by regression analysis, which indicates wall thinning with advancing age. This finding aligns with existing literature, suggesting that vascular remodelling is associated with ageing and potentially increases vulnerability to conditions such as aortic dissection [17]. Importantly,

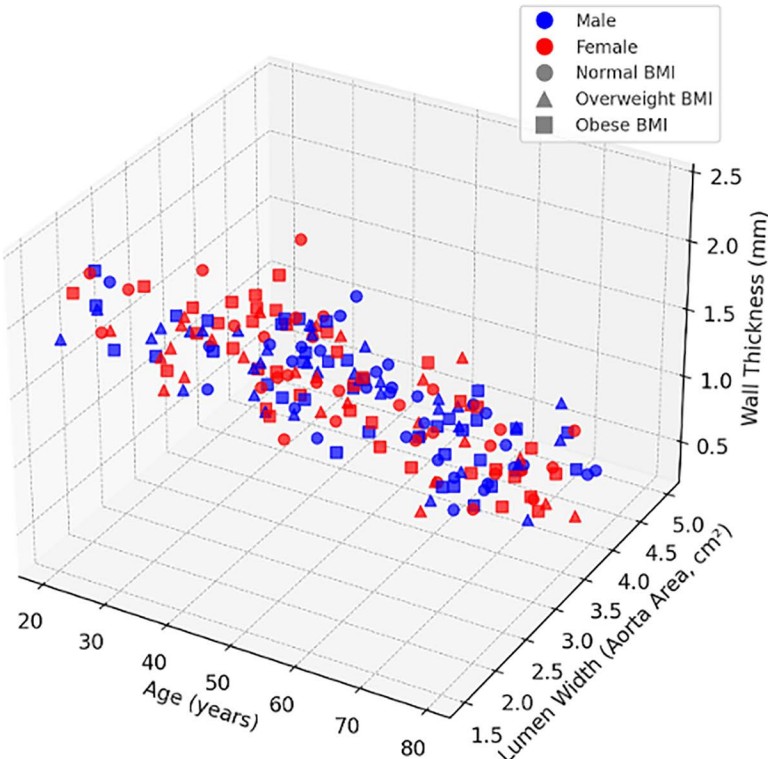

**Fig 7. A 3D scatter plot illustrates the relationship between age, gender, BMI, and their effect on the lumen width (aorta area) and wall thickness of the aorta.** The x-axis shows age (in years), the y-axis displays lumen width (in cm²), and the z-axis indicates wall thickness (in mm). Blue dots represent males, while red dots represent females. Different symbols represent BMI categories: circles for a normal BMI, triangles for an overweight BMI, and squares for an obese BMI.

we observed a convergence of wall thickness measurements between genders in individuals over 50 years, with younger males demonstrating significantly thicker aortic walls than females. This gender disparity decreases notably with age, corroborating previous observations in studies of vascular morphology [15,18].

BMI was significantly associated with increased wall thickness, exhibiting a significant positive correlation. Elevated BMI was consistently associated with increased aortic wall thickness, particularly pronounced among obese participants, with minimal gender differences observed in this subgroup. This finding underscores the critical role of obesity as a determinant of vascular remodelling, aligning well with established associations between increased BMI and adverse cardiovascular outcomes [19,20]. Conversely, lumen area showed an inverse relationship with wall thickness, indicating that a larger lumen typically corresponds to thinner aortic walls. This pattern was consistently observed across genders, although males displayed marginally thicker walls for comparable lumen area, highlighting subtle yet potentially clinically relevant gender-specific structural variations [17]. The observed gender differences in wall thickness may be attributed to hormonal factors influencing vascular structure, such as the protective effects of estrogen in females. Additionally, increased BMI is known to promote vascular remodelling through elevated hemodynamic stress and inflammatory pathways, which could explain the thicker walls observed in obese individuals.

Interestingly, conventional measurements of aortic diameter did not correlate significantly with wall thickness, challenging the prevailing assumptions in the vascular imaging literature that diameter alone reliably reflects overall vascular health [21]. Our novel approach, utilising lumen width measurements obtained from axial CT imaging, provided a more robust assessment of aortic structural integrity, emphasising the limitations inherent in diameter-based metrics [22].

Our findings both reinforce and add nuance to existing knowledge. The observed age-related decline in wall thickness aligns with previous imaging studies employing ultrasound and CT modalities, which also noted progressive aortic wall thinning with age [15,18]. However, the convergence of gender observed in older individuals adds a critical layer of complexity, suggesting that age-related physiological changes may reduce structural gender differences, a phenomenon supported by recent vascular ageing research [11]. The positive association between BMI and aortic wall thickness is consistent with previous findings [19,20], which linked obesity to increased arterial wall thickness, highlighting BMI as a crucial determinant that transcends gender differences. Extending these findings, our data indicate heightened gender disparities in wall thickness among individuals with higher BMI levels, particularly within overweight and obese populations, aligning with similar observations reported in a previous study [12].

The absence of correlation between traditional aortic diameter measurements and wall thickness, as observed in our study, contrasts with earlier suggestions that diameter is a reliable indicator of vascular health [23]. Our findings, consistent with those of Huang et al., highlight modality-specific and population-specific factors as potential explanations for discrepancies in aortic dimension assessments across studies [16]. Importantly, our novel lumen area metric demonstrates enhanced sensitivity, potentially capturing clinically significant structural alterations that may be missed by diameter-based measurements alone [16]. Clinically, these findings have significant implications for medical practice in Saudi Arabia. Establishing population-specific normative data for abdominal aortic dimensions significantly enhances diagnostic accuracy, enabling clinicians to distinguish between normal physiological variations and pathological conditions, such as abdominal aortic aneurysms (AAA). The convergence of gender-specific measurements with age underscores the need for age-adjusted diagnostic thresholds, particularly in younger demographics where gender disparities are most pronounced [24].

The robust correlation identified between BMI and aortic wall thickness highlights obesity as a modifiable risk factor. It emphasises the importance of targeted public health interventions aimed at obesity prevention and management to reduce the prevalence of cardiovascular disease in Saudi Arabia, a country currently experiencing rising obesity rates [19]. Furthermore, our lumen-focused measurement approach could enhance the early detection of vascular abnormalities, thereby promoting timely intervention strategies [16]. Our study's innovative pixel-to-pixel distance measurement technique demonstrated high precision in capturing subtle variations in aortic dimensions, suggesting a promising alternative to traditional measurement methods. Incorporating this approach into routine clinical imaging protocols could significantly enhance the assessment and monitoring of aortic health, particularly in populations with distinctive demographic profiles [25].

Our findings are especially relevant for clinical screening and risk stratification of abdominal aortic aneurysms (AAA). Currently, AAA diagnosis relies mainly on an arbitrary diameter threshold (≥3.0 cm). However, our results indicate that incorporating wall thickness and lumen area may provide a more accurate and individualised risk assessment. Despite its strengths, the study has limitations. It is cross-sectional and limited in terms of ethnic and socioeconomic diversity compared to the broader Saudi population, with a relatively modest sample size (n = 347), which may restrict generalizability. As this study was conducted in a single tertiary care centre, our findings may be affected by referral or selection bias. Although participants were chosen based on the absence of overt vascular pathology, they may not reflect the broader Saudi population, especially those without access to advanced imaging centres.

Additionally, while the chosen age range (20–80 years) provides valuable data, it does not include paediatric and very elderly populations. Moreover, focusing solely on healthy individuals means that our findings may not be applicable to pathological conditions, such as AAAs. Although MDCT offers excellent spatial resolution, its use is limited by radiation exposure and cost. Long-term data and comparisons with ultrasound and MRI would improve validation. Future research should incorporate other risk factors, such as blood pressure, smoking, and genetics, and examine longitudinal changes to understand vascular ageing processes better. Lastly, the potential differences in aortic measurements across various imaging methods (e.g., ultrasound versus MDCT) need further comparative studies [21].

Future research should validate and extend these findings through more extensive and diverse cohorts that include pathological cases, thereby enhancing the clinical applicability of our normative data. Longitudinal studies could further illuminate the dynamic relationship between aortic measurements, ageing, and BMI. Investigating the influence of additional factors, such as genetic predispositions and lifestyle variables, could refine our understanding of vascular health within the Saudi population. Finally, integrating the lumen width measurement technique into automated diagnostic systems could enhance clinical utility, promote early detection, and facilitate personalised treatment strategies [22].

## Conclusion

This study presents crucial normative data on abdominal aortic measurements in a healthy Saudi adult population, highlighting significant associations with age, gender, and BMI. The identified inverse correlation between age and wall thickness, the direct association with BMI, and the novel inverse relationship with lumen area provide critical insights into vascular health. The demonstrated limitations of traditional diameter measurements and the efficacy of our innovative lumen width approach underscore the necessity for advanced measurement techniques in clinical practice. Ultimately, these findings support the development of population-specific diagnostic guidelines, enhanced risk stratification, and improved clinical outcomes within Saudi Arabia and potentially worldwide.

## Author contributions

**Conceptualization:** Mohammad Ahmad Mostafa Alloush.

**Data curation:** Mohammad Ahmad Mostafa Alloush, Rashed Ali Alhamed.

**Formal analysis:** Mohammad Ahmad Mostafa Alloush, Rashed Ali Alhamed.

**Investigation:** Mazin Babikir Hassib, Rafat S. Mohtasib, Reham Mukhlid Almutairi.

**Methodology:** Husain Alturkistani.

**Project administration:** Rafat S. Mohtasib, Mohammed J. Alsaadi.

**Resources:** Husain Alturkistani, Reham Mukhlid Almutairi.

**Software:** Mazin Babikir Hassib.

**Supervision:** Husain Alturkistani, Rafat S. Mohtasib, Mohammed J. Alsaadi.

**Validation:** Reham Mukhlid Almutairi, Mohammed J. Alsaadi.

**Visualization:** Mazin Babikir Hassib, Rashed Ali Alhamed, Mohammed J. Alsaadi.

**Writing – original draft:** Mohammed J. Alsaadi.

**Writing – review & editing:** Mohammed J. Alsaadi.

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
