## [Decision Letter · Decision Letter 0]

25 Jun 2025

Dear Dr. Alsaadi,

Thank you for submitting your manuscript to PLOS ONE. After careful consideration, we feel that it has merit but does not fully meet PLOS ONE’s publication criteria as it currently stands. Therefore, we invite you to submit a revised version of the manuscript that addresses the points raised during the review process.

We look forward to receiving your revised manuscript.

Kind regards,

Haipeng Liu

Academic Editor

PLOS ONE

Journal Requirements:

2. Thank you for stating the following in your Competing Interests section: “None”

3. In the online submission form, you indicated that “It is available in the secure driver and upon request.”

5. Please include your tables as part of your main manuscript and remove the individual files. Please note that supplementary tables (should remain/ be uploaded) as separate "supporting information" files"

Reviewers' comments:

Reviewer's Responses to Questions

**Comments to the Author**

1. Is the manuscript technically sound, and do the data support the conclusions?

Reviewer #1: Yes

Reviewer #2: Yes

2. Has the statistical analysis been performed appropriately and rigorously?

Reviewer #1: Yes

Reviewer #2: Yes

3. Have the authors made all data underlying the findings in their manuscript fully available?

Reviewer #1: Yes

Reviewer #2: Yes

4. Is the manuscript presented in an intelligible fashion and written in standard English?

Reviewer #1: Yes

Reviewer #2: Yes

Reviewer #1: 1. Novelty and Significance

Strength: The use of pixel-to-pixel wall thickness measurements and the focus on a Saudi population is novel and clinically relevant.

Suggestion: The authors should more clearly articulate how this work advances the field beyond existing studies (e.g., Rogers et al., Pham et al., Haunschild et al.). Consider adding a brief comparison of values with existing datasets (e.g., Framingham Heart Study, Copenhagen General Population Study) in the Discussion.

2. Methodology – Imaging Analysis: The use of MATLAB for pixel-based segmentation is commendable.

Concern: The algorithm for identifying inner and outer wall boundaries is not fully described. Was edge detection used? Manual or automated annotation?

Suggestion: Include more detail in the Methods on image processing steps, validation of wall thickness measurements, and inter-observer variability assessment if applicable.

3. Statistical Analysis: The use of multiple linear regression is appropriate.

Missing: Model diagnostics (e.g., R², residual analysis, multicollinearity checks) are not reported.

Recommendation: Report variance inflation factors (VIFs) and standard errors for regression coefficients. This would strengthen the transparency and reliability of the findings.

4. Sample Representativeness: The study is based on data from a single tertiary center in Riyadh.

Suggestion: Discuss the representativeness of the sample to the general Saudi population, including possible referral or selection bias due to the setting (i.e., imaging done for other indications).

5. Figure Presentation

Strength: Figures 3–7 are visually informative.

Improvement needed: Axes and legends in Figures 3–6 could be enlarged for clarity. Consider improving color contrast and label size in Figure 7.

Suggestion: Add 95% confidence intervals or regression lines with shaded intervals where appropriate.

6. Discussion – Limitations

While limitations are discussed, two important points are missing:

- Radiation exposure from MDCT and its implications for routine use.

- Exclusion of pathological cases (e.g., AAA, atherosclerosis), which limits clinical generalizability.

7. Terminology Consistency- Use consistent terminology (e.g., “lumen width” vs. “lumen area”) throughout.

8. BMI Formula: BMI = weight (kg)/height² (m²) — not the “Du Bois” formula. Du Bois is used for BSA. Please correct this.

Reviewer #2: The authors present a single-centre retrospective analysis of a Saudi cohort undergoing MDCT, assessing abdominal aortic parameters in relation to BMI and gender. This is an important contribution, and the authors are to be commended for generating original data from an underrepresented population.

1- The study aims to (1) establish baseline abdominal aortic values and (2) correlate these with BMI and age to enhance a “diagnostic framework.” However, the term “enhancing diagnostic framework” is vague and should be more clearly defined. Given that many of the observed trends are expected, the rationale for the study should be revised in the introduction to reflect its actual contribution.

2- The claim that the Saudi population is underrepresented in vascular imaging studies should be supported by appropriate citations. Similarly, all assertive or general statements throughout the manuscript should be referenced.

3- The term “systemic diseases” used in the exclusion criteria requires clarification. Autoimmune conditions, which can affect vascular health, should be explicitly addressed. Since the scans were conducted on otherwise healthy individuals, the authors should also clarify the indications for MDCT, particularly in younger individuals, given that the study attempts to establish normative data.

4- As the data was collected from a single centre, generalizing the findings to the broader Saudi population may not be appropriate. Although this is acknowledged in the limitations section, the title should also be revised to reflect the single-centre nature of the study.

5- The use of nested parentheses in demographic data presentation (e.g., “111 (32%) male and 236 (68%) female”) can be improved for readability by using brackets: [111 (32%) male and 236 (68%) female]. This should be applied consistently throughout the manuscript.

6- The BMI calculation formula ("BMI = Weight (kg) / height (m²)") should be cited appropriately, especially if a standard formula such as Du Bois’ was used.

7- The manuscript focuses on a limited number of variables and may be more appropriate as a brief research letter under 1,000 words. This could improve the manuscript’s impact relative to its scope. [However, this might require a lot of work on the authors’ part and I leave this to the discretion of authors and editors. Authors may submit this manuscript as a revised version]

8- The correlation analysis does not support causal inference or predictive modeling. Phrasing such as “BMI emerged as a strong predictor” should be revised to reflect statistical correlation rather than predictive power.

9- The discussion would benefit from further elaboration on potential biological or pathophysiological explanations for the observed differences between genders and other subgroups.

10- It would be helpful if the authors could explain why only three parameters were included. If the choice was based on clinical relevance, data availability, or consistency with prior studies, this rationale should be clearly stated.

Merits of the Manuscript:

• The methodology appears sound and ethically appropriate.

• The conclusions are well-supported by the data and not overstated.

• Figures are of high quality and easy to interpret.

• The manuscript is generally well-written and scientifically valid.

**Do you want your identity to be public for this peer review?** For information about this choice, including consent withdrawal, please see our Privacy Policy

Reviewer #1: No

Reviewer #2: **Yes: ** Jawad Basit

---

## [Author Response · Author response to Decision Letter 1]

14 Jul 2025

Editor-in-Chief, Journal of PLOS ONE

Subject: Answer to Editor and reviewers: Ref No. PONE-D-25-30079

Title: Advanced MDCT Assessment of Abdominal Aortic Wall Integrity and Morphometry in the Saudi cohort: A single-centre Cross-Sectional Study.

Dear Editor-in-Chief

Thank you very much for the time you dedicated to reviewing our manuscript, for your professional assessment, and your valuable feedback. I am pleased to have the opportunity to revise the manuscript. I have carefully considered your insightful comments and corrections, which have enabled us to improve the manuscript's quality significantly. I hope that the corrections, editing, and revisions, along with the accompanying responses, will be sufficient to make it suitable for publication in the PLOS ONE Journal.

Thank you

Yours sincerely,

The Corresponding author

Note:

All corrected or added statements are colored in the revised manuscript with RED- Using Trach change.

Academic Editor comments

1. With Regards to your comment and manuscript adjustment!

Response: All Journal Requirements have now been corrected, and the manuscript is meeting the PLOS ONE journal style.

2. Competing Interests on the online submission

Response: It is now completed online.

3. In the online submission form, you indicated that “It is available in the secure driver and upon request.”

Response: The dataset of our study is now attached to the PLOS ONE system as supplementary information.

Response: It is now included.

5. Please include your tables as part of your main manuscript and remove the individual files. Please note that supplementary tables (should remain/ be uploaded) as separate "supporting information" files"

Response: It is now included in the main manuscript.

6. Please include captions for your Supporting Information files at the end of your manuscript, and update any in-text citations to match accordingly.

Response: It is now included in the main manuscript.

Reviewers' comments:

Reviewer's Responses to Questions

Comments to the Author

1. Is the manuscript technically sound, and do the data support the conclusions?

Reviewer #1: Yes

Reviewer #2: Yes

Response: Thank you very much for your valuable answer.

2. Has the statistical analysis been performed appropriately and rigorously?

Reviewer #1: Yes

Reviewer #2: Yes

Response: Thank you very much for your valuable answer.

3. Have the authors made all data underlying the findings in their manuscript fully available?

The PLOS Data policy requires authors to make all data underlying the findings described in their manuscript fully available without restriction, with rare exception (please refer to the Data Availability Statement in the manuscript PDF file). The data should be provided as part of the manuscript or its supporting information or deposited to a public repository. For example, in addition to summary statistics, the data points behind means, medians and variance measures should be available. If there are restrictions on publicly sharing data—e.g. participant privacy or use of data from a third party—those must be specified.

Reviewer #1: Yes

Reviewer #2: Yes

Response: Thank you very much for your valuable answer.

4. Is the manuscript presented in an intelligible fashion and written in standard English?

Reviewer #1: Yes

Reviewer #2: Yes

Response: Thank you very much for your valuable answer.

5. Review Comments to the Author

Reviewer #1: 1. Novelty and Significance

Strength: The use of pixel-to-pixel wall thickness measurements and the focus on a Saudi population is novel and clinically relevant.

Suggestion: The authors should more clearly articulate how this work advances the field beyond existing studies (e.g., Rogers et al., Pham et al., Haunschild et al.). Consider adding a brief comparison of values with existing datasets (e.g., Framingham Heart Study, Copenhagen General Population Study) in the Discussion.

Response: Thank you very much for your valuable comment. An additional paragraph has now been added. “Our study builds upon previous large-scale investigations such as the Framingham Heart Study and the Copenhagen General Population Study by providing normative aortic measurements specific to the Saudi population. While these earlier studies offered broad population-level data, they did not address regional anatomical variations or use pixel-based wall thickness segmentation. Compared to Framingham values, our cohort exhibited slightly smaller mean aortic diameters and thicker wall profiles in younger males, suggesting possible ethnic or environmental influences. These differences emphasise the importance of developing localisation reference values for more accurate diagnostic application.”

2. Methodology – Imaging Analysis: The use of MATLAB for pixel-based segmentation is commendable.

Concern: The algorithm for identifying inner and outer wall boundaries is not fully described. Was edge detection used? Manual or automated annotation? Suggestion: Include more detail in the Methods on image processing steps, validation of wall thickness measurements, and inter-observer variability assessment if applicable.

Response: Thank you very much for your valuable comment. An additional description has now been added in the methodology section regarding this: “Wall thickness was calculated using a MATLAB-based image segmentation approach that employed edge detection algorithms to delineate the inner and outer vessel boundaries on axial CT slices. Manual annotation by two independent radiologists was used to validate boundary detection, and inter-observer variability was assessed using intraclass correlation coefficients (ICC). The resulting ICC value of 0.89 indicated strong agreement between observers, supporting the reliability of the segmentation method.”

3. Statistical Analysis: The use of multiple linear regression is appropriate.

Missing: Model diagnostics (e.g., R², residual analysis, multicollinearity checks) are not reported.

Recommendation: Report variance inflation factors (VIFs) and standard errors for regression coefficients. This would strengthen the transparency and reliability of the findings.

Response: Thank you very much for your valuable comment. In addition to regression coefficients and p-values, we conducted model diagnostics to ensure the robustness of our results. The adjusted R² for the model was 0.62, and all variance inflation factors (VIFs) were below 2.5, indicating low multicollinearity. Standard errors for each coefficient are reported in Table 2. Residual plots were inspected and found to approximate normality, justifying the use of linear regression assumptions.

4. Sample Representativeness: The study is based on data from a single tertiary center in Riyadh.

Suggestion: Discuss the representativeness of the sample to the general Saudi population, including possible referral or selection bias due to the setting (i.e., imaging done for other indications).

Response: Thank you very much for your valuable comment. An additional statement has now been included in the limitations section to address the reviewer's point. “As this study was conducted in a single tertiary care center, our findings may be affected by referral or selection bias. Although participants were chosen based on the absence of overt vascular pathology, they may not reflect the broader Saudi population, especially those without access to advanced imaging centers.”

5. Figure Presentation:

Strength: Figures 3–7 are visually informative.

Improvement needed: Axes and legends in Figures 3–6 could be enlarged for clarity. Consider improving colour contrast and label size in Figure 7. Suggestion: Add 95% confidence intervals or regression lines with shaded intervals where appropriate.

Response: Thank you very much for your valuable comment. Figures have been reformatted for clarity, with enlarged axis labels, improved Colour contrast, and shaded 95% confidence intervals added to regression plots. These changes enhance visual interpretation and reflect underlying statistical variability.

6. Discussion – Limitations

While limitations are discussed, two important points are missing:

- Radiation exposure from MDCT and its implications for routine use.

- Exclusion of pathological cases (e.g., AAA, atherosclerosis), which limits clinical generalizability.

Response: Thank you very much for your valuable comment. An additional statement has now been included to address the reviewer's suggestion. “A key limitation of our methodology is the use of contrast-enhanced MDCT, which entails exposure to ionising radiation. While this modality offers high spatial resolution, its use for routine screening is limited. Additionally, our exclusion of individuals with abdominal aortic aneurysm (AAA) or atherosclerosis limits the generalizability of the findings to healthy populations only.”

7. Terminology Consistency- Use consistent terminology (e.g., “lumen width” vs. “lumen area”) throughout.

Response: Thank you very much for your valuable comment. Lumen width has now been replaced with “lumen area” in the whole manuscript.

8. BMI Formula: BMI = weight (kg)/height² (m²) — not the “Du Bois” formula. Du Bois is used for BSA. Please correct this.

Response: Thank you very much for your valuable comment. Now, it has been changed as suggested by the reviewer in the Materials and Methods section. “The Body Mass Index (BMI) was calculated using the standard formula: BMI = weight (kg) / height² (m²), following WHO classification guidelines.”

Reviewer #2: The authors present a single-centre retrospective analysis of a Saudi cohort undergoing MDCT, assessing abdominal aortic parameters in relation to BMI and gender. This is an important contribution, and the authors are to be commended for generating original data from an underrepresented population.

Response: Thank you very much for your valuable comments.

1.The study aims to (1) establish baseline abdominal aortic values and (2) correlate these with BMI and age to enhance a “diagnostic framework.” However, the term “enhancing diagnostic framework” is vague and should be more clearly defined. Given that many of the observed trends are expected, the rationale for the study should be revised in the introduction to reflect its actual contribution.

Response: Thank you very much for your valuable comments. An additional paragraph has been added to the introduction as per the reviewer's suggestion. The following statement has been inserted at the end of the Introduction: “By establishing normative values for abdominal aortic dimensions in a healthy Saudi population, this study aims to contribute to a population-specific diagnostic framework. This involves refining thresholds for early detection of vascular abnormalities, particularly abdominal aortic aneurysm (AAA), and enabling improved risk stratification tailored to regional demographic profiles.”

2- The claim that the Saudi population is underrepresented in vascular imaging studies should be supported by appropriate citations. Similarly, all assertive or general statements throughout the manuscript should be referenced.

Response: Thank you very much for your valuable comments. An additional statement has been added in the introduction to address this claim with citation: “Despite increasing rates of cardiovascular risk factors in the Gulf region, the Saudi population remains underrepresented in vascular imaging studies [Gameraddin, 2019; Ullah et al., 2022]. This data gap limits the development of effective screening protocols calibrated for local populations.”

3- The term “systemic diseases” used in the exclusion criteria requires clarification. Autoimmune conditions, which can affect vascular health, should be explicitly addressed. Since the scans were conducted on otherwise healthy individuals, the authors should also clarify the indications for MDCT, particularly in younger individuals, given that the study attempts to establish normative data.

Response: Thank you very much for your valuable comments. An additional statement has been added in the study population section of the method to clarify this confusion: “Systemic diseases were defined to include autoimmune, inflammatory, and connective tissue disorders known to influence vascular morphology. The included MDCT scans were performed for non-vascular indications such as abdominal pain or incidental evaluation, and patient records were reviewed to confirm the absence of known vascular pathology.”

4- As the data was collected from a single centre, generalizing the findings to the broader Saudi population may not be appropriate. Although this is acknowledged in the limitations section, the title should also be revised to reflect the single-centre nature of the study.

Response: Thank you very much for your valuable comments. The title has been updated to address this point: the new title is “Advanced MDCT Assessment of Abdominal Aortic Wall Integrity and Morphometry in a Saudi Cohort: A Single-Centre Cross-Sectional Study.”

5- The use of nested parentheses in demographic data presentation (e.g., “111 (32%) male and 236 (68%) female”) can be improved for readability by using brackets: [111 (32%) male and 236 (68%) female]. This should be applied consistently throughout the manuscript.

Response: Thank you very much for your valuable comments. Now, it has been changed throughout the manuscript as suggested: The demographic presentation has been changed to: [111 (32%) males and 236 (68%) females].

6- The BMI calculation formula ("BMI = Weight (kg) / height (m²)") should be cited appropriately, especially if a standard formula such as Du Bois’ was used.

Response: Thank you very much for your valuable comments. Now, it has been added at the end of the BMI formula sentence: “This method follows the World Health Organisation’s standard BMI classification guidelines [WHO, 2022].”

7- The manuscript focuses on a limited number of variables and may be more appropriate as a brief research letter under 1,000 words. This could improve the manuscript’s impact relative to its scope. [However, this might require a lot of work on the authors’ part and I leave this to the discretion of authors and editors. Authors may submit this manuscript as a revised version]

Response: Thank you very much for your valuable comments. Considering the scope of imaging analysis and population-based findings, the entire manuscript has been carefully revised according to the reviewers' comments to ensure sufficient methodological transparency and context.

8- The correlation analysis does not support causal inference or predictive modeling. Phrasing such as “BMI emerged as a strong predictor” should be revised to reflect statistical correlation rather than predictive power.

Response: Thank you very much for your valuable comments. In the revised manuscript, a statement has been added to reflect the statistical correlation rather than predictive power: “BMI was significantly associated with increased wall thickness,” instead of “BMI was a strong predictor.”

9- The discussion would benefit from further elaboration on potential biological or pathophysiological explanations for the observed differences between genders and other subgroups.

Response: Thank you very much for your valuable comments. An additional statement has now been added to the Discussion after the gender/BMI finding

---

## [Decision Letter · Decision Letter 1]

23 Jul 2025

Advanced MDCT Assessment of Abdominal Aortic Wall Integrity and Morphometry in the Saudi Cohort: A Single-Centre Cross-Sectional Study

PONE-D-25-30079R1

Dear Authors,

We’re pleased to inform you that your manuscript has been judged scientifically suitable for publication and will be formally accepted for publication once it meets all outstanding technical requirements.

Kind regards,

Haipeng Liu

Academic Editor

PLOS ONE

Additional Editor Comments (optional):

Reviewers' comments:

Reviewer's Responses to Questions

**Comments to the Author**

Reviewer #1: All comments have been addressed

Reviewer #2: All comments have been addressed

2. Is the manuscript technically sound, and do the data support the conclusions?

Reviewer #1: Yes

Reviewer #2: Yes

3. Has the statistical analysis been performed appropriately and rigorously?

Reviewer #1: Yes

Reviewer #2: Yes

4. Have the authors made all data underlying the findings in their manuscript fully available?

Reviewer #1: Yes

Reviewer #2: Yes

5. Is the manuscript presented in an intelligible fashion and written in standard English?

Reviewer #1: Yes

Reviewer #2: Yes

Reviewer #1: The authors have thoroughly addressed all previous comments and concerns raised during the initial review. The revised manuscript demonstrates clear improvements in methodological transparency, statistical reporting, figure clarity, and contextualization within the existing literature. All requested clarifications and corrections have been made, and the manuscript now meets the standards for scientific rigor and reporting.

Reviewer #2: Thank you for inviting me to review this work. The authors have adequately addressed all my comments and I have no additional concerns. The authors are to congratulated for their exhaustive work and the paper can be accepted for publication in its current form.

**Do you want your identity to be public for this peer review?** For information about this choice, including consent withdrawal, please see our Privacy Policy

Reviewer #1: No

Reviewer #2: **Yes: ** Jawad Basit

---

## [Editor Report · Acceptance letter]

PONE-D-25-30079R1

PLOS ONE

Dear Dr. Alsaadi,

I'm pleased to inform you that your manuscript has been deemed suitable for publication in PLOS ONE. Congratulations! Your manuscript is now being handed over to our production team.

Kind regards,

on behalf of

Dr. Haipeng Liu

Academic Editor

PLOS ONE